# Characterizing the Fungal Microbiome in Date (*Phoenix dactylifera*) Fruit Pulp and Peel from Early Development to Harvest

**DOI:** 10.3390/microorganisms8050641

**Published:** 2020-04-28

**Authors:** Edoardo Piombo, Ahmed Abdelfattah, Yaara Danino, Shoshana Salim, Oleg Feygenberg, Davide Spadaro, Michael Wisniewski, Samir Droby

**Affiliations:** 1Department of Agricultural, Forestry and Food Sciences (DISAFA), University of Turin, 10095 Grugliasco, TO, Italy; 2AGROINNOVA—Centre of Competence for the Innovation in the Agro-environmental Sector, University of Turin, 10095 Grugliasco, TO, Italy; 3Institute of Environmental Biotechnology, Graz University of Technology, 8010 Graz, Austria; 4Department of Ecology, Environment and Plant Sciences, Stockholm University, 10691 Stockholm, Sweden; 5Department of Postharvest Science, Agricultural Research Organization (ARO), The Volcani Center, Bet Dagan 5020000, Israel; 6U.S. Department of Agriculture—Agricultural Research Service (USDA-ARS), Kearneysville, WV 25430, USA

**Keywords:** date, *Phoenix dactylifera*, microbiome, metagenome, ITS, post-harvest

## Abstract

Date palm (*Phoenix dactylifera*) is considered to be a highly important food crop in several African and Middle Eastern countries due to its nutritional value and health-promoting properties. Microbial contamination of dates has been of concern to consumers, but very few works have analyzed in detail the microbial load of the different parts of date fruit. In the present work, we characterized the fungal communities of date fruit using a metagenomic approach, analyzing the data for differences between microbial populations residing in the pulp and peel of “Medjool” dates at the different stages of fruit development. The results revealed that *Penicillium*, *Cladosporium*, *Aspergillus*, and *Alternaria* were the most abundant genera in both parts of the fruit, however, the distribution of taxa among the time points and tissue types (peel vs. pulp) was very diverse. *Penicillium* was more abundant in the pulp at the green developmental stage (Kimri), while *Aspergillus* was more frequent in the peel at the brown developmental stage (Tamer). The highest abundance of *Alternaria* was detected at the earliest sampled stage of fruit development (Hababauk stage). *Cladosporium* had a high level of abundance in peel tissues at the Hababauk and yellow (Khalal) stages. Regarding the yeast community, the abundance of *Candida* remained stable up until the Khalal stage, but exhibited a dramatic increase in abundance at the Tamer stage in peel tissues, while the level of *Metschnikowia*, a genus containing several species with postharvest biocontrol activity, exhibited no significant differences between the two tissue types or stages of fruit development. This work constitutes a comprehensive metagenomic analysis of the fungal microbiome of date fruits, and has identified changes in the composition of the fungal microbiome in peel and pulp tissues at the different stages of fruit development. Notably, this study has also characterized the endophytic fungal microbiome present in pulp tissues of dates.

## 1. Introduction

Date palm (*Phoenix dactylifera*) is considered to be a highy important food crop in several African and Middle Eastern countries due to its nutritional value and health-promoting properties. The fruit serves as a source of several minerals, vitamins, carbohydrates, and fiber, and is consumed on a regular basis. Annual production in the Middle East was estimated at 5.1 million tons [1], and world production has been estimated at 7.2 million tons produced on an area of 11.2 million hectares, of which 11% are destined for export [1].

Among the date palm cultivars, “Medjool” is one of two of the most widely grown varieties within the U.S, Africa, and the Middle East. The other major variety is “Deglet Noor”, however, “Medjool” fruits are larger, softer, and sweeter [1]. Ripe “Medjool” date fruits usually drop to the bottom of a net covering each fruit cluster since ripening is spread over a period of several weeks starting in late August and ending in October, depending on the region and climate. Dates in the nets are collected every few days and then transferred to a packhouse. Some unripe fruits can be collected and allowed to ripen further by exposing them to the sun [2]. Dates can be consumed as a fresh fruit after harvest (Rutab) or semi-dry or dry (Tamer).

Dates go through a series of treatments after harvest to prepare them for marketing or long-term storage. First, unripe and damaged fruit are removed. This is followed by thermal or chemical disinfection of insect pests, heat drying (if necessary), cleaning with water on a packing line, grading for size, and lastly packaging. Challenges in the postharvest treatments of dates include identifying appropriate packaging technologies and the management of food safety issues, the latter of which includes contamination with mycotoxigenic fungi (e.g., *Aspergillus flavus*, *A. niger*, and *Penicillium chrysogenum* [3,4], as well as contamination with human foodborne pathogens, such as *A. fumigatus* [5]). 

Information on the level of microbial contamination of dates is very limited and only a few reports on this subject are present in the literature. The microbial load in date fruit is dependent on water activity (a*_w_*). Navarro [2] indicated, however, that *Aspergillus* species are able to grow at an a*_w_* slightly higher than 0.65. Al-Bulushi et al. [6] characterized the microbial composition of two popular date cultivars (‘Burny’ and ‘Kheniz’) produced in Oman using a metagenomic approach. They characterized the fungal species associated with different fruit parts at the Tamer stage and reported that the fungi common to both cultivars were Ascomycota (94%), Chytridiomycota (4%) and Zygomycota (2%). A total of 54 fungal species were detected, including species within the genera *Penicillium*, *Alternaria*, *Cladosporium*, and *Aspergillus*, which comprised more than 60% of the fungal operational taxanomic units (OTUs). Some potentially mycotoxin-producing fungi were detected in the stored dates, including *Aspergillus flavus*, *A. versicolor*, and *Penicillium citrinum*, but their relative abundance was very low (<0.5%). No differences were found in the fungal communities of the different parts (skin and pulp) [6]. 

Using standard culture-based methods, Shenasi and colleagues [3] determined the microbial load of 25 varieties of fresh dates and those stored at a high relative humidity. The main finding of this study was that the total microbial counts were high at the first stage of maturation (Kimri) and increased sharply at the next maturity stage (Rutab), then they decreased significantly at the final dried stage of maturation (Tamer). Aflatoxins were detected in 12% of the samples, although aflatoxigenic *Aspergillus* were detected in 40% of the varieties examined, exclusively at the Kimri stage of development. Additionally, lactic acid bacteria were present on fruits at the Rutab stage in some varieties, including all varieties in which aflatoxins or aflatoxigenic *Aspergillus* spp. were detected [3].

Hamad et al. [7] reported the microbial load of Rutab from different date cultivars grown in the Gulf Region and in Saudi Arabia. They found that all the fruit samples of six cultivars tested were contaminated with aerobic mesophilic bacteria at loads in the order 10^2^ to 10^5^ cfu/cm^2^. Notably, all samples were also found to be contaminated with coliform bacteria and the food-poisoning bacterium, *Staphylococcus aureus*, along with varying numbers of molds and yeasts. The mycotoxigenic fungi *A. flavus* and *A. parasiticus* were isolated only from a portion of the samples. The authors suggested that the levels of contamination with bacteria, molds, and yeasts were indicative of the general microbiological quality of the fruit and provided information that would affect the expected shelf life of the fruit [7]. 

To date, no studies have been conducted characterizing the microbiota of “Medjool” date fruit at the different stages of development and after ripening. The purpose of the present study was to characterize the composition of the fungal mycobiome (filamentous fungi and yeasts) in peel and pulp fruit tissues during fruit development in the orchard and in ripe fruit after harvest. We were able to demonstrate that a plethora of fungi can colonize date fruit at the different developmental stages, which provides information that can be used in the development of methods that address food safety. More specifically, *Alternaria* and *Cladosporium* were identified as the main fungal genera present at the beginning of fruit development (Hababauk), while the abundance of *Penicillium* became more significant after the green stage (Kimri), and *Cladosporium* was the dominant genus in the yellow stage (Khalal) and *Aspergillus* spp. were the most abundant in the brown stage (Tamer). These data indicate that the colonization and proliferation of fruit tissues by *Aspergillus* occurs over the course of fruit maturation. Furthermore, the presence of several endophytic genera were identified, which could serve as a potential source of biocontrol antagonists for the management of external and internal fungal contamination. 

## 2. Materials and Methods

### 2.1. Fruit Sampling

“Medjool” (*Phoenix dactylifera*) dates, the most common variety grown in Israel, were sampled in the southern Arava region, Israel (Yotvata), starting from 28 April 2017 to 4 September 2017 from approximately ten-years-old trees. Dates were sampled at the different stages of ripening: the first samples were taken at the Hababauk stage (28 April 2017, fruit set, with an average fruit weight of 0.37 g), the second at the Kimri stage (8 June 2017, with an average fruit weight of 12.5 g), the third at the Khalal stage (3 August 2017, with an average fruit weight of 21/0 g) and the fourth at the Tamer stage (4 September 2017, with an average fruit weight of 22.5 g) (Figure 1).

To determine the fruit quality parameters (moisture content and total soluble solids), 500 g of ripe fruit was homogenized with a blender and 5 g of the mixture was spread on a metal plate and inserted into a moisture analyzer (HC 103, Metler Toledo LLC, Columbus, OH, USA). To determine the sugar content (total soluble solids), 18 mL of distilled water was added to a 100 mL Erlenmeyer containing 2 g of the blended fruit tissue and shaken for 60 min on an orbital shaker set at 160 rpm. A drop of the supernatant was then placed on a 101-digital refractometer PR (ATAGO, Fukaya, Japan) to determine the percentage of sugars in the sample, with each measurement being the average of three replicates.

### 2.2. DNA Extraction

The peel and pulp of 20 fruits were removed aseptically, placed in Falcon 50 mL tubes and kept at −18 °C until use. The peel and pulp tissues were not separated in the samples collected at the Hababauk stage as the fruits were too small to effectively divide the peel and pulp. The tissues were freeze-dried, and each sample was ground in liquid nitrogen. Then, 40 mg of the powder was placed in sterile 1 mL Eppendorf tubes and subjected to DNA extraction using a Wizard © Genomic DNA Purification kit (Promega Corporation, Madison, WI, USA), following the manufacturer’s protocol. The DNA concentration and quality in each sample was determined using a nanodrop ND-1000 spectrophotometer (Thermo Fisher Scientific, Waltham, MA, USA).

### 2.3. Library Preparation and Metagenomic Sequencing

The total DNA concentration in each sample was adjusted to 5.0 ng μL−1. The fungal ITS2 region was amplified using the universal primers ITS3/KYO2 and ITS4 [8]. All primers were modified to include Illumina adapters (www.illumina.com). PCR reactions were conducted in a total volume of 25 μL containing 12.5 μL of KAPA HiFi HotStart ReadyMix (Kapa Biosystems, Wilmington, MA, USA), 1.0 μL of each primer (10 μM), 2.5 μL of DNA template, and 8.0 μL of nuclease-free water. The amplifications were conducted in a T100 thermal cycler (Bio-Rad) using the following protocol: 3 min at 98 °C followed by 30 cycles of 30 s at 95 °C, 30 s at 50 °C, and 30 s at 72 °C. All amplifications ended with a final extension of 1 min at 72 °C. Nuclease-free water (QIAGEN, Valencia, CA, USA) was substituted for the template DNA in the negative controls. All amplicons and amplification mixtures, including negative controls, were sequenced on a MiSeq platform using V3 chemistry (Illumina, San Diego, CA, USA).

### 2.4. Data Analysis

Trimmomatic 0.36 was used for the primer and adapter trimming using the following parameters: LEADING:3 TRAILING:3 SLIDINGWINDOW:4:15 MINLEN:100 [9]. Paired-end reads were assembled using PANDAseq 2.11 using a minimum overlapping of 40bp and an alignment threshold of 0.9 [10]. Chimeric sequences were identified and filtered using VSEARCH 1.4.0 [11]. The usable reads for the operational taxonomic unit (OTU) clustering in each sample were calculated and samples with less than 1000 usable reads were filtered out. The UCLUST algorithm [12] of the software package QIIME 1.9.1 [13] was used to cluster the sequences at a similarity threshold of 97% against the UNITE dynamic database released on 12 January2017 [14]. Sequences that failed to cluster against the database were de novo clustered using the same algorithm. The most abundant sequence in each OTU was selected as a representative sequence for the taxonomic assignment using the BLAST algorithm [15] as implemented in QIIME 1.9.1.

The OTU table was normalized by rarefaction to an even sequencing depth in order to remove sample heterogeneity. The rarefied OTU table was used to calculate the alpha diversity indices including the Observed Species (Sobs), Chao1, and Shannon metrics. MetagenomeSeq’s cumulative sum scaling (CSS) was used as a normalization method for other downstream analyses, including the taxa relative abundance, alpha diversity, and group significance [16]. Alpha diversities were compared based on a two-sample *t*-test using non-parametric (Monte Carlo) methods and 999 Monte Carlo permutations (999). The CSS normalized OTU table was analyzed using Bray–Curtis metrics [17] and utilized to evaluate the beta-diversity and construct PCoA (Principal Coordinates Analysis) plots [18]. The volatility analysis [19] was performed with QIIME2 v. 2019.1.0, after the CSS normalization of the data.

The differentially represented genera were evaluated with edgeR v. 3.8 [20], using 0.05 as the significance threshold for the FDR (False Discovery Rate) (Appendix A). The function “glmQLFTest” was used for the testing.

## 3. Results

The results of the analysis of the maturity and ripening parameters of the dates at different stages (Figure 1) is presented in Table 1. As expected, moisture decreased as the fruit development proceeded, while sugar content increased.

The high throughput sequencing resulted in 6,169,409 paired-end ITS (Internal Transcribed Spacer) reads. After trimming, paired-end assembly, and quality and chimera filtering, 4,054,098 full-length ITS reads with an average length of 349 bp were kept. These reads were clustered into ITS OTUs at a 97% similarity threshold. The number of reads and OTUs varied between samples (Appendix A). Nontarget (non-fungal) reads were filtered and only fungal sequences were retained for the downstream analysis, amounting to 3043 fungal OTUs.

### 3.1. Diversity

A principal coordinates analysis clustering of the samples indicated clustering of the tissue type and developmental stage (Figure 2; Appendix A). Clearly evident clustering of the Hababauk, Kimri pulp, Khalal peel and Tamer peel samples was observed based on the tissue type and developmental stage, while the samples of the Kimri peel, Khalal pulp, and Tamer pulp exhibited a higher degree of variability, with some samples clustering with samples of other stages. The Shannon index and the observed OTUs were calculated as alpha diversity metrics (Table 2; Figure 3). Both the metrics indicated no significant difference in the alpha diversity of the considered stages and tissues.

### 3.2. Taxonomy

Ascomycota was the dominant phylum across all samples, followed by Basidiomycota. The percentage of Basidiomycota in the peel was greater than in the pulp at all developmental stages and exhibited an increase over time, starting from less than 5% at the Hababauk stage and reaching 24.88% in the peel at the Tamer stage (Figure 4). Chytridiomycota and Glomeromycota (approximately 0.02% and 0.08%, respectively) were also present, although as a relatively small percentage. The phylum Mucoromycota (approximately 0.79%) was also detected, especially in the peel at the Kimri stage.

Appendix A contains the quantification across the various tissues and development stages of every genus for which OTUs were obtained in this study. *Penicillium* was the most prevalent genus (averaging 24.22%), amounting to more than 92% of the total microbiota in the pulp at the Kimri stage. Other genera comprising more than 1% of the microbiome on average were *Cladosporium* (19.99%), *Aspergillus* (12.18%), *Alternaria* (9.72%), *Filobasidium* (6.34%), *Sarocladium* (5.8%), *Purpureocillium* (4.00%), *Issatchenkia* (2.64%), *Malassezia* (1.69%), *Phaeotheca* (1.45%) and *Aureobasidium* (1.4%) (Figure 5; Appendix A).

Three of the four most abundant genera were primarily represented by a single species, except for *Alternaria*, which was represented by two species (Appendix A). Thus, the most abundant species in the date pulp and peel samples were *Penicillium aethiopicum* (23.70% of the total mycobiome on average), *Cladosporium herbarum* (18.70%), *Aspergillus tubingensis* (12.10%), *Alternaria subcucurbitae* (5.09%), and *Alternaria betae-kenyensis* (4.41%). Importantly, however, the taxonomic assignment of OTUs beyond the genus level can be problematic with ITS. Fourteen OTUs were present in all sample types (tissue type and developmental stage) (Table 3). Three were unidentified, and the other 11 were assigned to the following genera: *Sarocladium*, *Aspergillus*, *Filobasidium*, *Alternaria*, *Issatchenkia*, *Penicillium*, *Cladosporium*, *Aureobasidium*, and *Metschnikowia*.

*Cladosporium* and *Alternaria* were the most prevelant genera in the Hababauk samples, while *Penicillium* and *Aspergillus* spp. reached maximum abundance in the following stages. Kimri peel samples did not have a genus that could be definitively considered as the most abundant. The most abundant genus in the peel samples at the Khalal stage was *Cladosporium* (65%), while *Aspergillus* spp. were the most abundant genus at the Tamer stage (62.6%). Indeed, *Aspergillus* species are xerophilous and grow at a low moisture and *a_w_* with a high sugar content, typical conditions of the Tamer stage [2]. *Penicillium* spp. and *Alternaria* spp. made up respectively 6% and 5% of the peel fungal microbiome, and their presence did not significantly change throughout the study. Regarding the yeast community, *Candida* spp. were significantly more abundant in the peel samples at the Tamer stage than in the previous stages, increasing from 0.01% to 1% of the fungal microbiome. The same pattern was observed for *Naganishia* spp. (0.01% to 2.67%) and *Cutaneotrichosporon* spp. (0.00% to 1.27%), while the maximum presence of *Rhodotorula* spp. in the peel was found at the Khalal stage (3.57%) and then it decreased to zero. *Phaeoteca* spp., on the contrary, were only present in the peel at the Kimri stage, where they represented a significant portion of the microbiome (10.17%). The abundance of *Metschnikowia* spp., known for their biocontrol potential against postharvest pathogens [21,22,23], was very variable within the samples, and no significant differences were observed between the stages or tissue types, even though it made up 4.82% of the fungal peel microbiome at the Kimri stage and only 0.2% at the Tamer stage. The most abundant species of the previously listed genera were *Candida tropicalis*, *Naganishia albida*, *Cutaneotrichosporon dermatis*, *Rhodotorula mucilaginosa*, *Phaeoteca triangularis*, and *Metschnikowia pulcherrima*.

The microbiome of the pulp was different from that of the peel. *Penicillium* represented the greatest fraction of the fungal microbiome of the pulp samples at the Kimri stage (92%), but subsequently it decreased and was 45.2% of the total at the Khalal stage and 12.3% at the Tamer one. *Aspergillus*, *Alternaria*, and *Cladosporium*, on the other hand, represented 5.3%, 1.7%, and 9.5% of the pulp microbiome on average, respectively, and their presence did not change significantly over the course of the fruit development. The yeast population in the pulp, however, was very low, compared to the peel population. The abundance of *Candida* spp. and *Metschnikowia* spp. were always very low in the pulp (0.04% and 0.10% on average) samples.

## 4. Discussion

The analysis of the beta-diversity (Figure 2) of the mycobiome at the different developmental stages of date fruit revealed a distinct and unique microbiota in each developmental stage, indicating a dynamic shift rather than a stable community of fungi. The most abundant fungal genera were *Penicillium*, *Cladosporium*, *Aspergillus*, and *Alternaria*. This correlates well with the study of Al-bulushi et al. [6], indicating that these genera comprised more than 60% of the mycobiome in other date cultivars. *Alternaria* and *Cladosporium*, hygrophilous genera growing at high *a_w_* levels, were the most abundant genera at the start of ripening (Hababauk stage), and notably were also reported to represent a significant fraction of the date leaf mycobiome by Ben Chobba et al. [24]. This similarity suggests that the date fruit mycobiome at the start of development may be similar to the mycobiome of leaves in date trees. The most abundant species of the genera *Alternaria* and *Cladosporium* were *Alternaria betae-keyensis*, *Alternasia subcucurbitae*, and *Cladosporium herbarum*. *Alternaria* is a genus known for the production of both host-specific and unspecific mycotoxins [25], but neither *A. betae-kenyensis* nor *A. subcucurbitae* have so far undergone genome sequencing or a thorough metabolomic analysis, so it is difficult to speculate about their role or effect in *Phoenix dactylifera* microbiomes. *Cladosporium herbarum* is mostly known as epiphyte on leaves, but also for its production of allergens [26,27,28,29]. However, dates are never consumed at this stage of development, so its presence does not pose a health risk.

### 4.1. Peel

None of the detected genera were distinctively dominant in the peel at the Kimri stage. *Cladosporium* was the most present in the Khalal stage, while *Aspergillus* spp. constituted the majority of the fungal microbiome in the Tamer stage. This may indicate that *Cladosporium* and *Aspergillus* are more adaptable to high levels of sugars and possibly other nutrients present in the peel. Therefore, their numbers increased more rapidly than other fungi. The high presence of *Aspergillus* spp. at the Tamer stage suggests that these fungi are capable of withstanding the low moisture and high sugar content in date fruit at this phenological stage (Table 1). The most frequent *Aspergillus* sp. found in this study was *Aspergillus tubingensis*, a species belonging to the Nigri section. In the study of Palou et al. [30], the most frequent *Aspergillus* species isolated by “Medjool” dates, the same variety used in this work, belonged to the Nigri section, but the authors were unable to reach species identification, therefore our results correlate well with their findings. *A. tubingensis* was also the most abundant *Aspergillus* species reported on dates by Al-Bulushi et al. [6]. The genome of *A. tubingensis* contains clusters putatively able to produce anthracenone naphthacenedione compounds with immunosuppressive activity [31], but, even if it belongs to the section Nigri, it does not seem able to produce ochratoxin A or fumonisins, and on the contrary it has a putative ocratoxinase gene. This species, previously reported to control grey mold on tomatoes [32], should be investigated for its ability to outcompete mycotoxins producing *Aspergillus* spp. on dates, and its putative ability to degrade ochratoxins.

The yeast community was dynamic in the peel during the course of the fruit development. At the Kimri stage, when the fruit moisture was the highest (Table 1), the most abundant yeast was *Phaeoteca triangularis*, a species reported to be common in humidifiers, bathing facilities and other locations with high water activity [33]. At the Khalal stage, the most abundant species was *Rhodotorula mucilaginosa*, known for its biocontrol potential [34,35], and previously detected in dates by Al-bulushi and colleagues [6]. At the Tamer stage the most abundant yeasts belonged to the genera *Naganishia*, *Candida*, and *Cutaneotrichosporon*. In particular, the vast majority of the *Candida* population was constituted by *Candida tropicalis*, reported previously by Al-bulushi and colleagues [6]. The role and source of this yeast in dates is not clear and should be further studied since some strains are known to be human pathogens [36], while others have been used to control postharvest diseases in bananas, strawberries, and litchis [37,38,39]. Little information is available in the literature regarding *Naganishia albina* and *Cutaneotrichosporon dermatis*, the most abundant species of the respective genera.

### 4.2. Pulp

*Penicillium*, *Aspergillus*, *Cladosporium*, and *Alternaria* have been reported to be the genera most associated with mold growth in “Medjool” dates [30]. Surprisingly, however, stable populations of these genera were identified as endophytes in the current study throughout the fruit development. Although several studies have reported the existence of root endophytes in *P. dactylifera* [40,41,42], our work is the first to demonstrate the existence of fungal endophytes in the pulp of date fruit. At present, however, it is difficult to speculate on the role of these fungal endophytes in pulp tissues. They may represent symbiotic fungi that colonize fruit tissues early during flowering and fruit set. Some of these endophytic symbionts (e.g., *Aspergillus* sp.) may become pathogenic and cause internal black rot under conductive conditions [30]. Interestingly, *Penicillium* spp. completely dominated the endophytic date fungal microbiome at the Kimri stage, but then decreased in relative abundance as fruits further developed and reached maturity. One possible explanation could be that conditions were favorable for growth of this genus as the date fruit reached the Kimri stage, but then *Penicillium* spp. growth stopped or greatly diminished with further fruit develpment, leaving roughly the same level of abundance in a constantly increasing volume of fruit. This would result in a reduction in the relative biomass of *Penicillium* per gram of fruit. Alternatively, other fungi may have suppressed *Penicillium* growth as their relative abundance increased. *Penicillium aethiopicum*, the most abundant species of *Penicillium* found in this study, is able to produce viridicatumtoxin (tetracycline-like antibiotic), griseofulvin (antifungal activity), and tryptoquialanine (tremorgenic mycotoxin) [43,44]. In a previous study [6], the most common *Penicillium* sp. growing on dates was found to be *Penicillium griseofulvum*, another species able to produce griseofulvin. It is possible that the ability to synthetize this antifungal metabolite, whose putative cluster has been identified recently [45], is important for the colonization of dates by *Penicillium* spp.

The presence of non-pathogen endophytes in date fruits should be the object of future studies since certain endophytes can potentially be a safety hazard (production of mycotoxins), while others may be useful as beneficial antagonists competing with unfavorable epiphytic and endophytic fungi, or as a potential source of biologically active substances [46]. In this regard, a study of the endophytes present in the roots of date trees has already produced promising results. For example, the strain NAT001 of *Penicillium crustosum*, isolated from the roots of *P. dactylifera*, produces compounds with potent anti-proliferative and anti-inflammatory action, which can inhibit the growth and migration of certain cancer cell lines [47].

## 5. Conclusions

This work constitutes a comprehensive analysis of the fungal microbiome of date (*Phoenix dactylifera*) fruits, that included peel vs. pulp tissues at the different stages of fruit development. The present study was the first to document the presence of endophytes in date fruit pulp, which can serve as a baseline for future analyses. The study of endophytes is especially promising for the identification of strains that are able to promote growth, control diseases, or produce useful compounds.

## Figures and Tables

**Figure 1 microorganisms-08-00641-f001:**
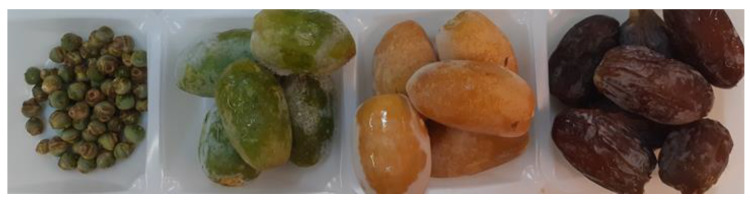
Photographs of dates at sequential stages of ripening. From left to right: Hababauk, Kimri or green fruit, Khalal or yellow fruit, Tamer or brown fruit.

**Figure 2 microorganisms-08-00641-f002:**
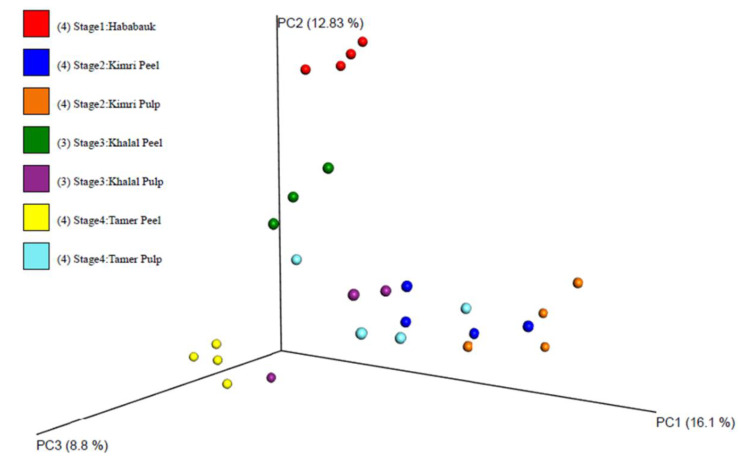
Clustering of the samples according to Bray–Curtis distance, after cumulative sum scaling (CSS) normalization.

**Figure 3 microorganisms-08-00641-f003:**
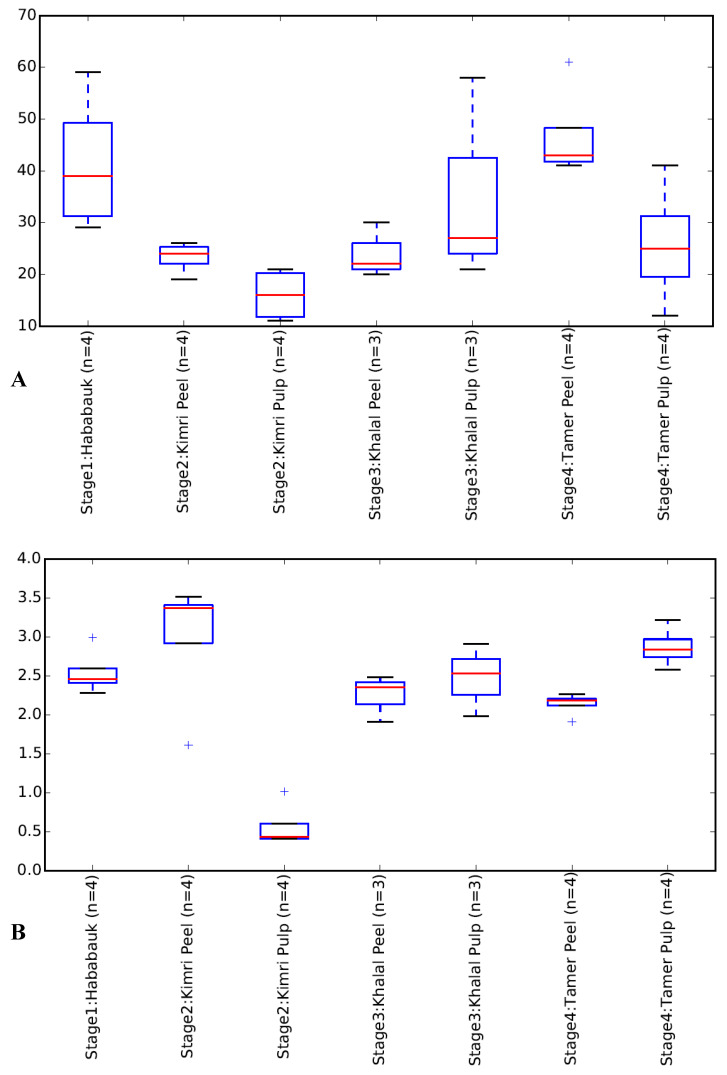
Shannon Diversity Index (**A**) and number of observed OTUs (**B**) of the analyzed tissues and developmental stages of dates.

**Figure 4 microorganisms-08-00641-f004:**
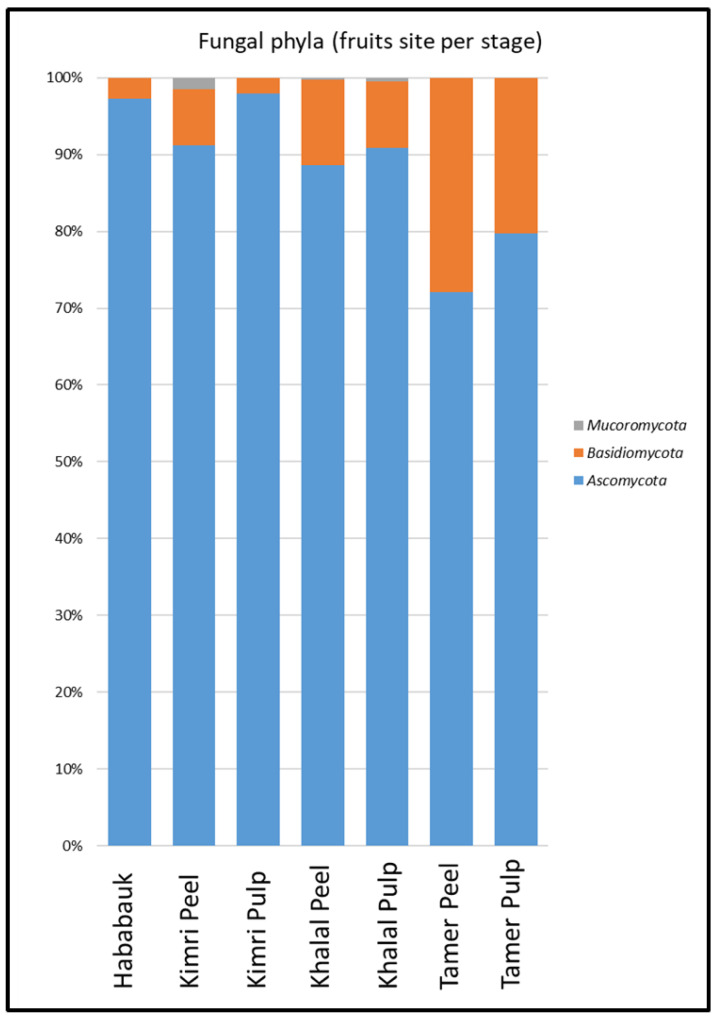
Percentages of the various phyla comprising the fungal microbiome of the different tissue types (peel and pulp) and developmental stages.

**Figure 5 microorganisms-08-00641-f005:**
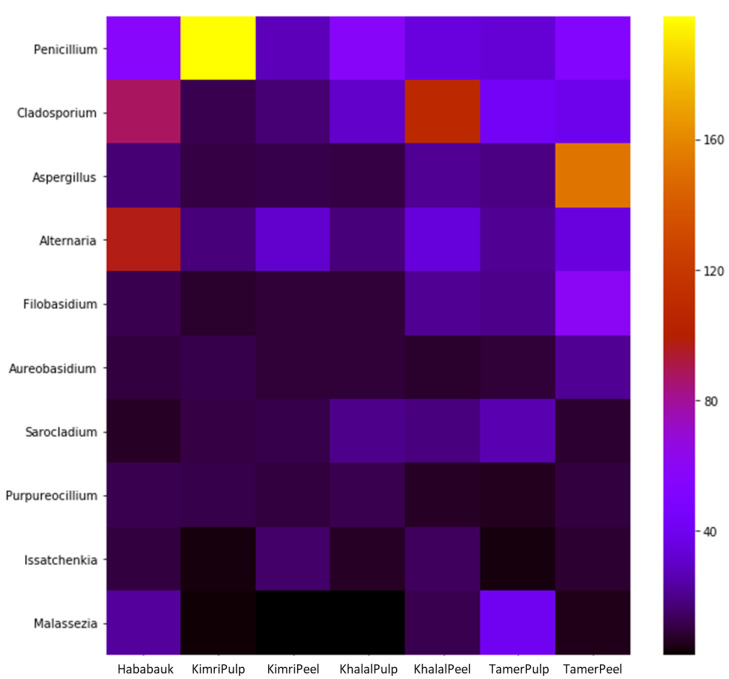
Heatmap of the CSS-normalized read count of the most abundant genera detected in the different tissue types and developmental stages of date fruit.

**Table 1 microorganisms-08-00641-t001:** Average moisture and total soluble solids measured in “Medjool” dates at the Hababuk, Kimri, Khalal, and Tamer stages of ripening.

Fruit Stage	Moisture (%)	Total Soluble Solids (%)
Hababauk	72.9 ± 1.2	6.1 ± 0.6
Kimri	84.2 ± 0.4	6.5 ± 1.0
Khalal	61.3 ± 0.4	27.0 ± 0.81
Tamer	26.8 ± 2.0	51.5 ± 0.6

**Table 2 microorganisms-08-00641-t002:** Alpha diversity comparison among the considered tissues and stages of the study. The selected alpha diversity metrics were the Shannon Diversity Index and the number of operational taxonomic units (OTUs).

Group1	Group2	*p*-Value Shannon Diversity Index	*p*-Value OTU Number
Stage3:Khalal Peel	Stage4:Tamer Pulp	0.161	0.891
Stage3:Khalal Peel	Stage1:Hababauk	0.503	0.290
Stage4:Tamer Peel	Stage2:Kimri Peel	0.292	0.179
Stage4:Tamer Pulp	Stage1:Hababauk	0.305	0.313
Stage4:Tamer Peel	Stage1:Hababauk	0.122	0.799
Stage3:Khalal Pulp	Stage1:Hababauk	0.860	0.752
Stage2:Kimri Pulp	Stage1:Hababauk	0.105	0.315
Stage2:Kimri Pulp	Stage2:Kimri Peel	0.168	0.231
Stage3:Khalal Peel	Stage2:Kimri Peel	0.357	0.893
Stage3:Khalal Peel	Stage2:Kimri Pulp	0.091	0.344
Stage3:Khalal Peel	Stage4:Tamer Peel	0.553	0.462
Stage3:Khalal Peel	Stage3:Khalal Pulp	0.460	0.714
Stage4:Tamer Pulp	Stage3:Khalal Pulp	0.363	0.717
Stage2:Kimri Pulp	Stage4:Tamer Peel	0.087	0.151
Stage3:Khalal Pulp	Stage4:Tamer Peel	0.308	0.489
Stage2:Kimri Pulp	Stage3:Khalal Pulp	0.105	0.196
Stage3:Khalal Pulp	Stage2:Kimri Peel	0.477	0.359
Stage4:Tamer Pulp	Stage2:Kimri Peel	0.888	0.784
Stage4:Tamer Pulp	Stage4:Tamer Peel	0.126	0.168
Stage4:Tamer Pulp	Stage2:Kimri Pulp	0.090	0.333
Stage2:Kimri Peel	Stage1:Hababauk	0.526	0.231

**Table 3 microorganisms-08-00641-t003:** Number of OTUs of each genus present in each and every one of the analyzed tissues and stages (Hababauk, Kimri Pulp, Kimri Peel, Khalal Pulp, Khalal Peel, Tamer Pulp, and Tamer Peel).

Genus	Core OTUs
*Penicillium*	1
*Aspergillus*	1
*Alternaria*	2
*Cladosporium*	2
*Issatchenkia*	1
*Filobasidium*	1
*Sarocladium*	1
*Metschnikowia*	1
*Aureobasidium*	1
Unidentified	3

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
