# Peer review of "Characterizing the Fungal Microbiome in Date (Phoenix dactylifera) Fruit Pulp and Peel from Early Development to Harvest"

_microorganisms, 2020, doi:10.3390/microorganisms8050641_

Round 1

Reviewer 1 Report

The manuscript is well written and follow a logical flow. The issue has been addressed well in in production. Methods are standard. There is a confusion between water activity and water content which they differ in concept. I have pointed this out in the attached pdf file. Results are presented well. The discussion part, well presents the ideas and discusses the results and uncertainties. 

Overal, after minor correcrions the manuscript is acceptable in presnt form. 

Author Response

I thank you for your comments, all the corrections indicated on the revised pdf were added to the paper, with one exception. At line 65 you asked if the water activity measurement was a percentage and suggested to add the percentage symbol, but aw is generally presented as a fraction of one, not a percentage, and therefore it does not need the symbol.

Regarding your comment about the measuring of water activity (lines 117-124), you were right, it was a mistake. Only moisture and total soluble solids were analyzed, and consequently these are the data shown in table 1. A small modification in the discussion (lines 239-240) was introduced to reflect our measurement of moisture and not water activity.

Other small modifications present in the paper are the results of the correction of grammatical errors suggested by the other reviewers.

Reviewer 2 Report

Minor revision

In this paper, authors characterized the fungal microbiome in pulp and peel od date fruit in different stages of the development.  The study is very interesting, relevant, well designed and presented.

After checking the text for grammatical errors, the paper may be accepted for publication.

Author Response

I thank you for your comments.

The manuscript has been checked and corrected.

Reviewer 3 Report

In the manuscript  the authors describe the microbial populations residing in date pulp vs. the peel of 'Medjool’ dates at different stages of fruit development. The methods used in the work are clearly presented and the results are supported by reliable statystical analyses. The findings reported show an interesting (not surprisingly) variety of fungal population in the different tissues at different ripening stages. Alternaria and Cladosporium were identified as the main fungal genera present at the beginning of fruit development (Hababauk), while the abundance of Penicillium became more significant after the green stage (Kimri), Cladosporium was the dominant genus in the yellow stage (Khalal) and Aspergillus spp. were the most abundant in the brown stage (Tamer). The data presented give informations possibly useful in the development of methods that address food safety and biological controls of fungal diseases in dates.

Author Response

I thank you for your comments.